# Engineering probiotics to inhibit *Clostridioides difficile* infection by dynamic regulation of intestinal metabolism

Elvin Koh [1,2,3,7], In Young Hwang[1,2,3,7], Hui Ling Lee[1,2,3], Ryan De Sotto[1,2,3], Jonathan Wei Jie Lee [1,2,4], Yung Seng Lee [1,2,5], John C. March[6] & Matthew Wook Chang [1,2,3 ✉]

*Clostridioides difficile* infection (CDI) results in significant morbidity and mortality in hospitalised patients. The pathogenesis of CDI is intrinsically related to the ability of *C. difficile* to shuffle between active vegetative cells and dormant endospores through the processes of germination and sporulation. Here, we hypothesise that dysregulation of microbiome-mediated bile salt metabolism contributes to CDI and that its alleviation can limit the pathogenesis of CDI. We engineer a genetic circuit harbouring a genetically encoded sensor, amplifier and actuator in probiotics to restore intestinal bile salt metabolism in response to antibiotic-induced microbiome dysbiosis. We demonstrate that the engineered probiotics limited the germination of endospores and the growth of vegetative cells of *C. difficile* in vitro and further significantly reduced CDI in model mice, as evidenced by a 100% survival rate and improved clinical outcomes. Our work presents an antimicrobial strategy that harnesses the host-pathogen microenvironment as the intervention target to limit the pathogenesis of infection.

[1] NUS Synthetic Biology for Clinical and Technological Innovation (SynCTI), National University of Singapore, Singapore, Singapore. [2] Synthetic Biology Translational Research Programme, Yong Loo Lin School of Medicine, National University of Singapore, Singapore, Singapore. [3] Department of Biochemistry, Yong Loo Lin School of Medicine, National University of Singapore, Singapore, Singapore. [4] Department of Medicine, Yong Loo Lin School of Medicine, National University of Singapore, Singapore, Singapore. [5] Department of Paediatrics, Yong Loo Lin School of Medicine, National University of Singapore, Singapore, Singapore. [6] Department of Biological and Environmental Engineering, Cornell University, Ithaca, NY, USA. [7] These authors contributed equally: Elvin Koh, In Young Hwang. ✉email: bchcmw@nus.edu.sg

*C*lostridioides (or *Clostridium*) *difficile* infection (CDI) is the leading cause of healthcare-associated infectious diarrhoea globally[1–3]. In the United States alone, CDI annually causes over 500,000 infections and 20,000 deaths, with an annual estimated healthcare cost of $4.8 billion[4]. The eradication of CDI is impaired by the frequent recurrence of the disease; 20.9% of CDI cases recur once or more within 30 days post treatment[5]. Paradoxically, while the CDI treatment regimen involves antibiotic therapy, antibiotics have been established as a major risk factor for CDI. Antibiotics trigger and prolong a state of dysbiosis in the intestine, characterised by a loss of bacterial diversity and altered production of microbiome-derived intestinal metabolites, hindering the recovery of the normal colonic microbiome and leading to CDI and recurrent CDI (rCDI)[1,4].

The pathogenesis of CDI and rCDI is intrinsically related to the ability of *C. difficile* to shuffle between active vegetative cells and dormant endospores through the processes of germination and sporulation[6]. Carriers of *C. difficile* endospores may remain asymptomatic as the endospores remain dormant in the human gastrointestinal tract. Furthermore, endospores are inherently resistant to antibiotics[6]. Perturbation to the native balance of the microbiome, or dysbiosis, is suggested to trigger the germination of *C. difficile* endospores and allow rapid expansion of *C. difficile*, leading to infection[7–9]. Upon colonisation of the gastrointestinal tract, vegetative *C. difficile* secretes exotoxins, such as *C. difficile* toxin A (TcdA) and B (TcdB), that impair tight junctions and disrupt the actin cytoskeleton in human intestinal epithelial cells[10,11]. The toxins result in extensive colonic inflammation and epithelial tissue damage, which manifest as symptomatic CDI[10]. The persistence of the dysbiosis induced by antibiotic therapy enables the residual endospores of *C. difficile* that evade treatment to germinate and lead to rCDI[4].

Recent evidence has pointed to a causal relationship between CDI and dysregulation of microbiome-mediated bile salt metabolism[7,12,13], which affects the germination of *C. difficile*[9,13]. In humans, bile salts are synthesised and conjugated in the liver and secreted into the upper intestine (duodenum), reabsorbed at the distal small intestine (ileum) and returned to the liver via the portal circulation[9,14]. Bile salts are discharged from the human liver as primary bile salts conjugated to taurine or glycine. The conjugated primary bile salts then undergo deconjugation and subsequent 7α-dehydroxylation to secondary bile acids by members of the intestinal microbiota[15–17]. As a result of these modifications, bile salts in the colon exist predominantly as unconjugated primary or secondary forms[9,15,16]. Physiological microbiome-mediated bile salt metabolism can be disrupted by antibiotic-induced dysbiosis, for instance, leading to an over-accumulation of conjugated bile salts such as taurocholate in the colon[9,13]. Taurocholate is a strong germinant of *C. difficile* routinely utilised to induce the germination of *C. difficile* endospores in vitro. Correspondingly, dysbiosis reduces the intestinal levels of unconjugated and secondary bile salts, which reportedly inhibit *C. difficile* colonisation[9,12,13,18].

As dysbiosis-mediated bile salt metabolism contributes to CDI, we, therefore, hypothesised that microbiome modulation of bile salt metabolism can be targeted to disrupt the pathogenesis of CDI. To test this hypothesis, we engineered probiotics to restore intestinal bile salt metabolism in response to antibiotic-induced microbiome dysbiosis (Fig. 1). We designed a genetic circuit harbouring a genetically encoded sensor, amplifier and actuator that enabled hydrolase-mediated modulation of the conjugated bile salts (potential germinants) to the unconjugated forms. This modulation was dynamically controlled by a genetically encoded sensor that detects sialic acid, a proxy signal for microbiome dysbiosis whose levels are elevated upon antibiotic-induced dysbiosis in the intestine[19]. We demonstrated that the engineered probiotics limited the germination of endospores and the growth of vegetative cells of *C. difficile* in vitro and, furthermore, significantly reduced CDI in murine models, as evidenced by a 100% survival rate and improved clinical symptoms. Our work presents a targeted microbiome-modulation strategy that harnesses the host–pathogen microenvironment as the intervention target to limit the pathogenesis of infection, suggesting the potential for the modulation of bile salt metabolism to serve as a mechanism of action for CDI therapy.

## Results

**Hydrolase-mediated bile salt deconjugation inhibits *C. difficile* germination and growth.** To validate our hypothesis that the modulation of bile salt profiles controls *C. difficile* germination, we assessed the effect of taurocholate and its deconjugated form, cholate, on the germination and growth of *C. difficile* (Supplementary Fig. 1A). While the physiological level of bile salts varies along the intestinal tract, we have taken 2 mM of bile salts as an average representative concentration to use for in vitro assays[9,20]. Figure 2A, B show that cholate significantly decreased the germination of the endospores by up to 81% and the number of viable vegetative cells of *C. difficile* strains by up to 82%. We then chose the bile salt hydrolase Cbh of *Clostridium perfringens* as an enzyme to be introduced into probiotics to deconjugate taurocholate into cholate. We used this enzyme because Cbh uses taurocholate as a preferred substrate and remains more functional at physiological pH than other bile salt hydrolases[21]. Figure 2C shows that recombinant Cbh (Supplementary Fig. 1B, C) converted 99.2% of the taurocholate into cholate. Cbh also deconjugated glycocholate, another conjugated primary bile salt present in the gut, with similar efficiency. Glycocholate is also reported to be a germinant of *C. difficile*, and the deconjugating activity of Cbh against both taurocholate and glycocholate achieved >99% reduction of conjugated bile salts (Fig. 2D). Figure 2E, F show that the group with recombinant Cbh exhibited a 96% lower endospore germination rate and 89% fewer viable vegetative cells than the control group with taurocholate and no Cbh. These results suggested that recombinantly expressed Cbh deconjugated taurocholate into cholate and that this enzymatic deconjugation significantly inhibited the germination of endospores and the growth of vegetative cells of *C. difficile*.

**Engineering of dysbiosis-sensing circuits in probiotics.** We chose *E. coli* Nissle 1917 as the probiotic host because it is an extensively studied probiotic with a long safety record in humans[22,23]. Furthermore, as a gram-negative bacterium, *E. coli* Nissle 1917 can be employed in combination with the current CDI therapy, which entails the use of antibiotics that target gram-positive bacteria. We used the previously reported auxotrophic *E. coli* Nissle 1917 strain as the base strain, which requires exogenous D-alanine for survival (denoted by EcN), because it allows antibiotic-free selection, enhances plasmid stability, and provides a means for biocontainment[22,24].

Based on increasing evidence that microbiome dysbiosis preludes the onset of CDI[7,8,12,13,19,25], we hypothesised that a genetic sensor that regulates the expression of Cbh in response to microbiome dysbiosis might increase the efficacy of our engineered probiotics against CDI. We chose sialic acid as a proxy signal for dysbiosis because the level of sialic acid is elevated in the intraluminal space upon antibiotic treatment, which has been postulated to be a result of a dysbiosis-mediated imbalance between sialic acid-catabolising and/or sialidase-expressing members of the microbiome[19]. Sialic acid also supports the pathogenesis of gastrointestinal infections, including CDI, possibly because pathogens can use sialic acid as a carbon

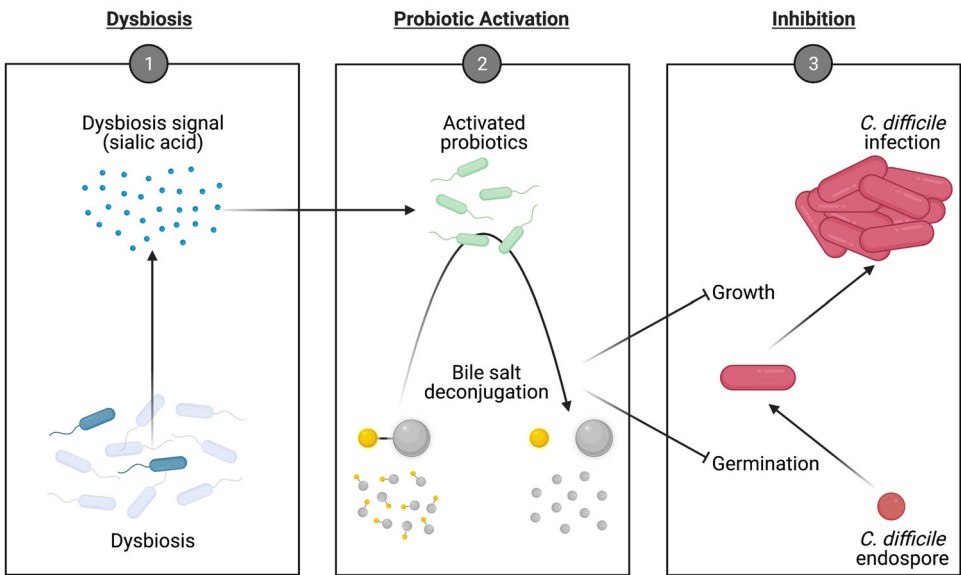

**Fig. 1 Schematic of the engineered probiotics against CDI.** Probiotics were engineered to restore intestinal bile salt metabolism in response to antibiotic-induced microbiome dysbiosis in order to inhibit the germination and growth of *C. difficile*.

source and decorate the cellular surface with sialic acid to evade host responses[19,26–28].

We selected the promoter pNanA as a core sensor element because it has been postulated to be responsive to sialic acid[29]. We characterised pNanA with the transcriptional regulator NanR (Supplementary Fig. 2A). Supplementary Fig. 2B shows that pNanA was responsive to sialic acid when NanR was expressed, while pNanA alone resulted in a high background with a low response. Then, to improve the dynamic range of the pNanA-NanR sensor (termed 'Sensor'), we modulated the expression level of NanR using a set of constitutive promoters and ribosome binding sites. Fig. 3A shows that the combination of J23113 (promoter) and rbs4 (RBS) led to the highest dynamic range in response to sialic acid. We also observed that the pNanA-NanR sensor dynamically regulated the expression of GFP (reporter protein), where the removal of sialic acid brought the expression back to the basal level (Supplementary Fig. 2C, D). Notably, the pNanA sequence contains a catabolite activator protein (CAP) binding site. In line with this feature, Fig. 3B shows that glucose significantly reduced the GFP expression in response to sialic acid. The level of glucose is significantly lower in the ileum, caecum and colon[30], where *C. difficile* primarily colonises[13]. Therefore, we hypothesised that the pNanA-NanR sensor can provide another layer of expression control such that the expression of Cbh increases when the engineered probiotics reach the target sites: the ileum, caecum and colon.

**Engineering of sensor-amplifier-actuator circuits in probiotics.** Next, we examined whether the expression of Cbh (termed 'Actuator'), when induced by the sialic acid sensor, could inhibit the germination of *C. difficile* endospores. Figure 3C shows a reduction in endospore germination by up to 47%, which indicated a significantly lower efficacy than that of purified recombinant Cbh (Fig. 2D). We also observed low conversion of taurocholate into cholate (Fig. 3D). Therefore, we hypothesised that the low germination inhibition might be due to insufficient expression of Cbh. To test this hypothesis, we added an amplifier module to the sensor-actuator circuit. Specifically, the transcriptional activator gene *cadC* was placed under the control of the promoter pNanA so that CadC (termed 'Amplifier') could

activate the promoter pCadBA to amplify Cbh expression. The amplifier module was evaluated for sialic acid-responsivity (Supplementary Fig. 2E). CadC regulates the cad operon and has been shown to be pH sensitive[31], providing an additional layer of expression control[32]. A module with constitutive GFP expression was also evaluated for comparison (Supplementary Fig. 2F). The final sensor-amplifier-actuator circuit, where Cbh served as an actuator (Fig. 3E), resulted in significantly increased actuator (Cbh) expression of the circuit (Fig. 3F). The expression level of Cbh under sialic acid induction was comparable to that of purified recombinant Cbh (Fig. 3G), which led to significant germination and growth inhibition (Fig. 2D, E). These results suggest that the EcN harbouring the aforementioned sensor-amplifier-actuator circuit (denoted by EcN-Cbh), which comprises pNanA, NanR, CadC, pCadBA and Cbh, might significantly inhibit the germination and growth of *C. difficile*.

**The engineered probiotics inhibit *C. difficile* germination and growth.** We then examined the extent to which EcN-Cbh could deconjugate taurocholate into cholate and reduce the germination of *C. difficile* endospores. In response to sialic acid, EcN-Cbh fully converted taurocholate into cholate (Fig. 4A) and caused a 98% reduction in endospore germination (Fig. 4B). Notably, the absence of sialic acid led to less but still considerable deconjugation (45%) and germination reduction (90%), likely due to the basal expression of Cbh in EcN-Cbh. Basal expression of Cbh did not significantly alter the growth of the host strains (Supplementary Fig. 3A). In addition, no extracellular deconjugation activity of EcN-Cbh was observed (Supplementary Fig. 4), suggesting that the deconjugation action of EcN-Cbh remained intracellular, although Cbh is reportedly secreted in gram-positive *C. perfringens*[33].

Next, we determined whether EcN-Cbh could inhibit the growth of the vegetative cells of *C. difficile*. Figure 4C shows that EcN-Cbh strongly reduced the number of viable vegetative cells. To further investigate the mechanism of this reduction, we evaluated the numbers of viable vegetative *C. difficile* cells after culture with EcN-Cbh, taurocholate and cholate. Supplementary Fig. 3B, C indicate that the growth inhibition was due to cholate. Figure 4C shows that 1-hour preincubation with taurocholate was

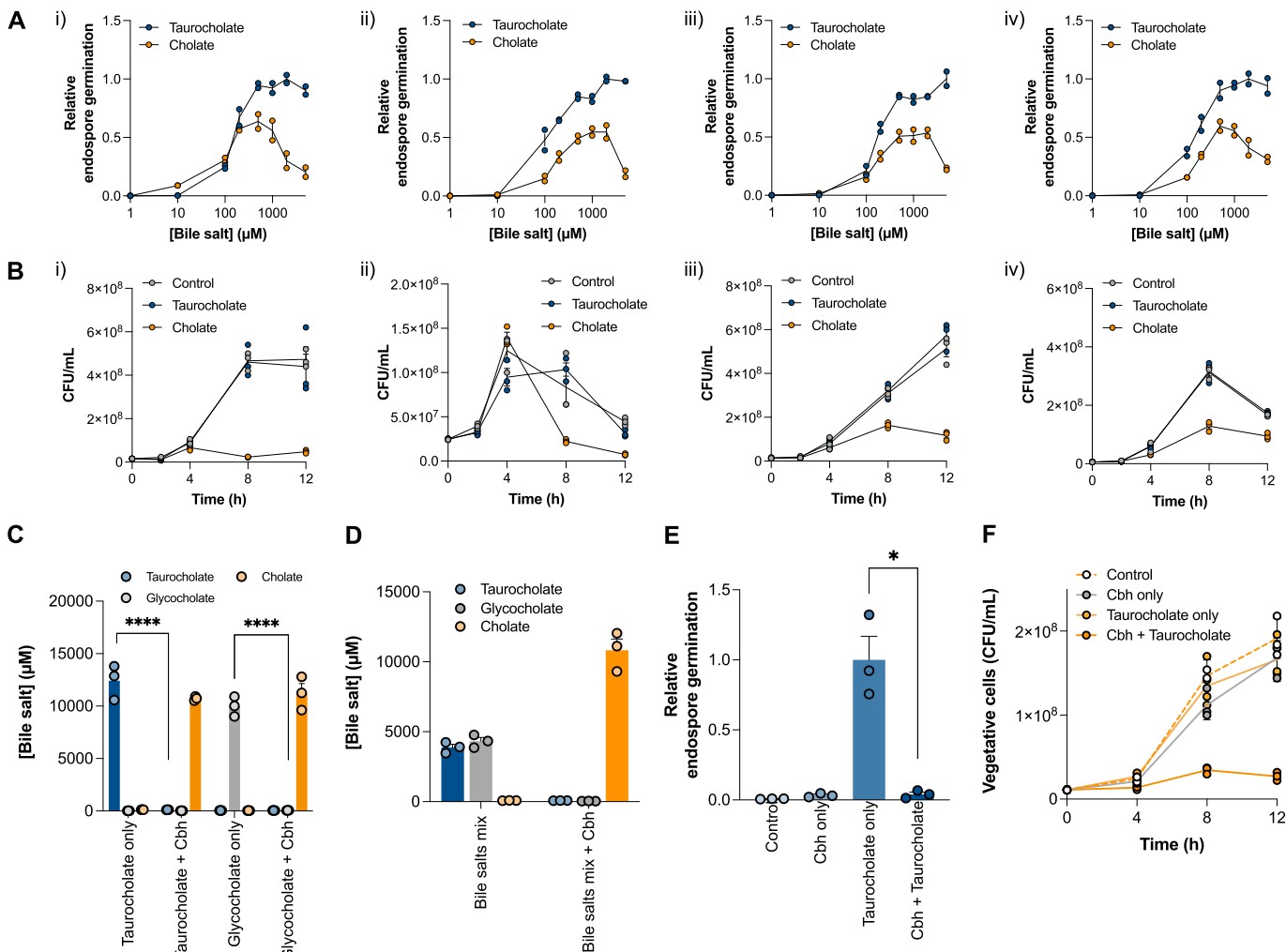

**Fig. 2 The bile salt hydrolase Cbh inhibits *C. difficile* via deconjugation of taurocholate. A** Germination of endospores from *C. difficile* strains ((i) CD630, (ii)VPI10463, (iii) BAA1870, and (iv) 9689), under different concentrations of bile salt (taurocholate or cholate). The experiments were set with the same initial endospore number and indicated the amount of germination as a percentage of the maximal germinated CFUs. $n = 2$ independent experiments. **B** Growth of vegetative cells from *C. difficile* with 2 mM bile salt (taurocholate or cholate) or without bile salt (control) in BHIS media. $n = 3$ independent experiments. **C** Quantification of bile salts following incubation with Cbh. The concentrations of taurocholate, glycocholate and cholate were determined through LC/MS. Unpaired two-sided Student's *t*-tests were performed to compare the bile salt concentrations between the Cbh-treated and control groups ($p < 0.0001$). ****$P < 0.0001$. $n = 3$ independent experiments. **D** Quantification of bile salts (5 mM taurocholate and 5 mM glycocholate) following incubation with Cbh. The concentrations of taurocholate, glycocholate and cholate were determined through LC/MS. $n = 3$ independent experiments. **E** Germination of purified *C. difficile* endospores and **F** growth of vegetative *C. difficile* following treatment with Cbh for 1 or 12 h, respectively. The reactions were set up as follows: the *control* group contained blank buffer, the *Cbh-only* group contained purified Cbh-his6 without bile salt (final concentration of 2 mM), the *taurocholate-only* group contained taurocholate without enzymes and the *Cbh + taurocholate* group contained purified Cbh and taurocholate. Relative endospore germination was calculated by normalising to taurocholate-only group. Unpaired two-sided Student's *t*-test was performed to compare the *Cbh + taurocholate* group and the *taurocholate-only* control group ($p = 0.0085$). *$P < 0.05$. $n = 3$ independent experiments. All data were presented as mean values with error bars representing SEMs of triplicates unless stated otherwise. Source data are provided as a Source Data file.

sufficient to significantly inhibit the vegetative growth of *C. difficile* cultured with EcN-Cbh. Together, these results suggest that EcN-Cbh significantly inhibits the germination of endospores and vegetative cells of *C. difficile*.

**The engineered probiotics reduce *C. difficile* toxicity**. To investigate whether and the extent to which germination and growth inhibition by EcN-Cbh could lead to a reduction in the toxicity of *C. difficile*, we first assessed the amount of the *C. difficile* exotoxin TcdA when *C. difficile* was cocultured with EcN-Cbh. Figure 4D shows that EcN-Cbh significantly decreased the level of TcdA in the culture, which likely resulted from lowered

germination and growth of *C. difficile*. Then, we evaluated the viability of human epithelial colorectal adenocarcinoma cells (Caco-2 cells) upon exposure to the supernatant of *C. difficile*. EcN-Cbh significantly improved the viability of Caco-2 cells exposed to *C. difficile* (Fig. 4E). These results together suggest that pre-treatment with EcN-Cbh reduced *C. difficile* toxicity.

**The engineered probiotics inhibit CDI in a murine model**. The aforementioned in vitro results prompted us to hypothesise that EcN-Cbh might inhibit CDI in vivo. To test this hypothesis, we evaluated whether and the extent to which EcN-Cbh could reduce mortality and morbidity in a murine model of CDI that was

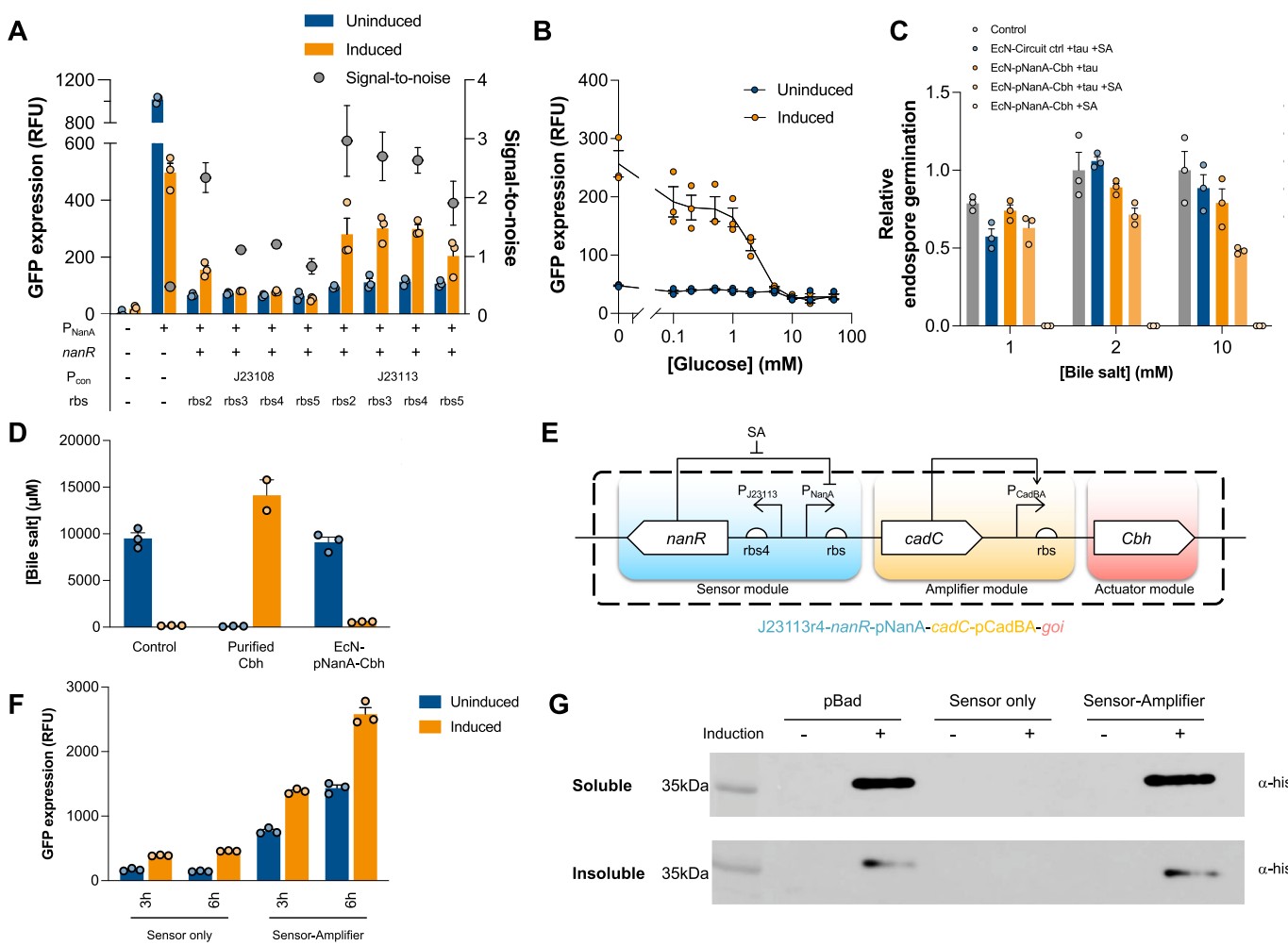

**Fig. 3 Development and characterisation of a sialic acid-responsive biosensor. A** Optimisation of pNanA induction via modulation of the nanR co-expression level with different combinations of constitutive promoters ($P_{con}$) and ribosome binding sites (rbs). GFP was expressed under the control of pNanA as a reporter gene in all constructs. Induction was performed with sialic acid. The relative GFP expression after 3 h of induction is shown. Signal-to-noise represents the ratio between the induced to the uninduced GFP expression. **B** Relative GFP fluorescence of EcN expressing the selected pCon-rbs combination (J23113-rbs4-*nanR*-pNanA-*gfp*) after 3 h of incubation with varying concentrations of glucose with or without 0.2% sialic acid. **C** Germination efficiency of purified *C. difficile* endospores in taurocholate and/or sialic acid treated with EcN expressing Cbh from a selected sialic acid sensor construct (J23113-*nanR*-pNanA-*cbh*; *EcN-pNanA-Cbh*). No-EcN reaction and circuit control (*EcN-circuit ctrl*) were used as controls. The circuit control expressed GFP in lieu of Cbh under the same construct design. The germinated endospore CFU values were normalised to the no-EcN control values. **D** Quantification of bile salts following incubation with EcN-Cbh from a selected sialic acid sensor construct (J23113-*nanR*-pNanA-*cbh*; *EcN-pNanA-Cbh*). No-Cbh reaction and purified Cbh reaction were used as controls. The concentrations of taurocholate and cholate were determined through HPLC. **E** Circuit design of the final sensor-amplifier construct (J23113-*nanR*-pNanA-*cadC*-pCadBA-*gfp*) consisting of sensor, amplifier, and actuator modules. **F** Relative GFP fluorescence of EcN expressing the *sensor-only* construct (J23113-*nanR*-pNanA-*gfp*) or the *sensor-amplifier* construct (J23113-*nanR*-pNanA-*cadC*-pCadBA-*gfp*). The relative GFP expression after 3 and 6 h of induction is shown. **G** Immunoblot of Cbh-his6 expressed from different constructs in EcN. The *pBad* construct (*araC*-pBad-*cbh*) was induced with ʟ-arabinose for 16 h, and the *sensor-only* construct (J23113r4-*nanR*-pNanA-*cbh*) and *sensor-amplifier* construct (J23113r4-*nanR*-pNanA-*cadC*-pCadBA-*cbh*) were induced with sialic acid for 16 h. S represents the soluble fraction, and IS represents the insoluble fraction. The expected size of Cbh-his6 is 38 kDa. The band on the ladder corresponds to 35 kDa. All data were presented as mean values with error bars representing SEMs of triplicates (*n* = 3 independent experiments). Source data are provided as a Source Data file.

previously established[34,35] (Fig. 5A). The model mice were given engineered probiotics prior to being exposed to a virulent strain of *C. difficile*, VPI10463, that has been reported to induce significant mortality and symptomatic displays[34,36]. The evaluation was conducted using a treatment group (EcN-Cbh) and five control groups: (i) a no-sensor control (EcN-Cbh-S⁻) group, (ii) a

no-amplifier control (EcN-Cbh-A⁻) group, (iii) a no-actuator control (EcN-Cbh⁻) group, (iv) a wild-type control (EcN-WT) group and (v) an infection control (CD) group (Fig. 5B). Figure 5C shows the extent of deconjugation activities of the probiotics. Supplementary Figs. 5, 6 and 8 show further characteristics of these probiotics, such as growth, expression,

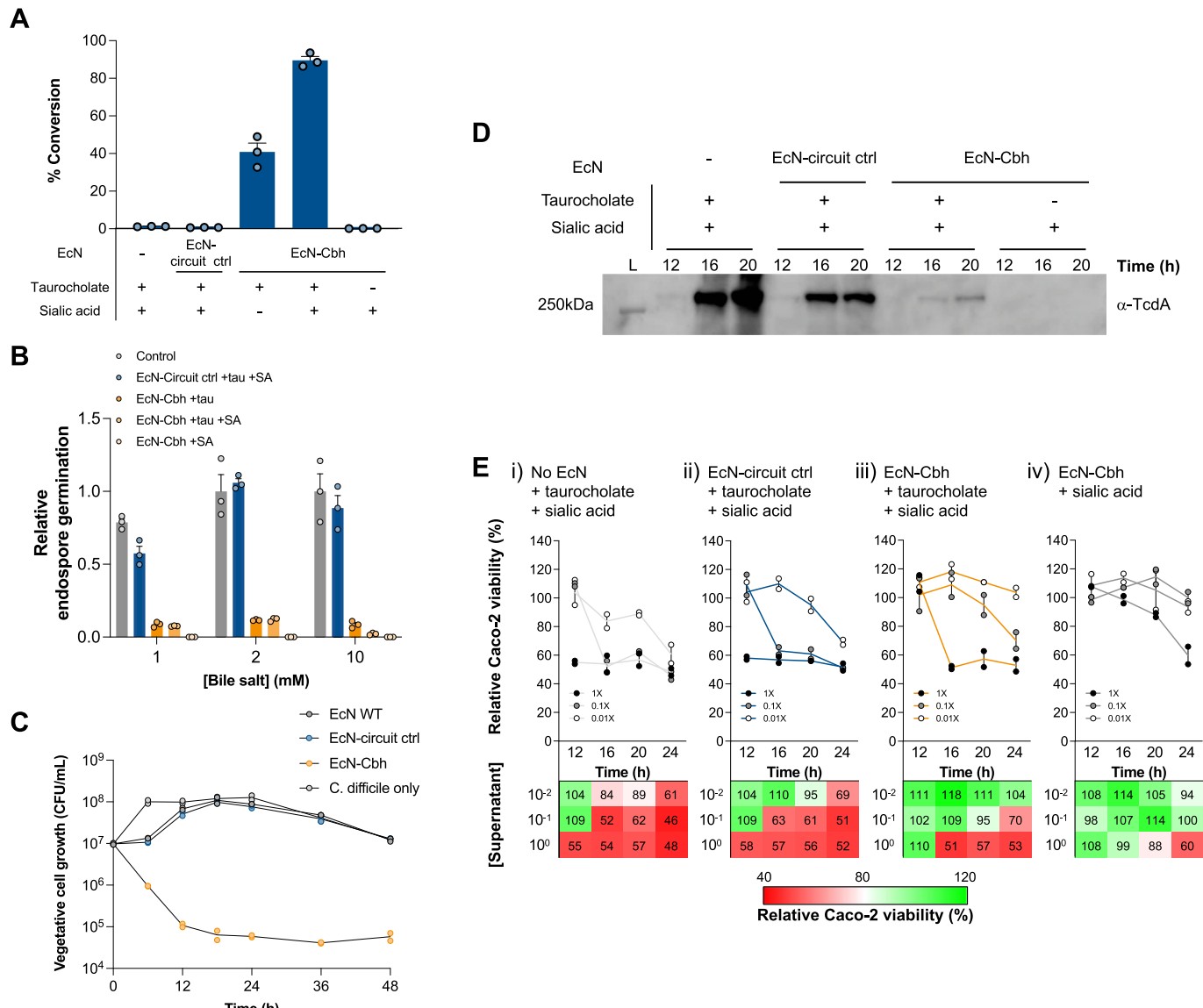

**Fig. 4 The engineered probiotics inhibit C. difficile endospore germination and growth and reduce C. difficile toxin secretion. A** Quantification of taurocholate and cholate after 3 h incubation of taurocholate and/or sialic acid with EcN or EcN expressing the sensor-amplifier-*cbh* construct (*EcN-Cbh*). Conversion efficiency (cholate/taurocholate) is shown. No-EcN reaction and circuit control (*EcN-circuit ctrl*) reaction were used as controls. The circuit control expressed GFP in lieu of Cbh under the same construct design. **B** Germination efficiency of purified *C. difficile* endospores in taurocholate (tau) and/ or sialic acid (SA) treated with EcN-Cbh. The germinated endospore CFU values were normalised to a no-EcN control group. Unpaired two-sided Student's *t*-test was performed to compare the EcN-Cbh and *EcN-circuit ctrl* groups. *$P < 0.05$. **C** CFU values from coculture assays between vegetative *C. difficile* cell cultures and wild-type EcN (*EcN WT*), *EcN-circuit ctrl*, or *EcN-Cbh* cultures preincubated for 1 h with taurocholate. $n = 2$ independent experiments. **D** Immunoblot of the *C. difficile* toxin TcdA from concentrated supernatants at different time points of germinating *C. difficile* culture. Reactions with or without taurocholate were treated with EcN-Cbh and then incubated with *C. difficile* endospores. No-EcN reaction and *EcN-circuit ctrl* reaction were used as controls. EcN was induced with taurocholate and/or sialic acid. The blot was probed with an anti-tcdA antibody. L represents the protein ladder. The expected size of TcdA is 308 kDa. **E** Relative cell viability of Caco-2 cells treated with supernatants collected from germinating *C. difficile* cultures under the conditions outlined for Fig. 4D (i) no-EcN control, (ii) *EcN-circuit ctrl*, (iii) EcN-Cbh with taurocholate, and (iv) EcN-Cbh without taurocholate. The supernatants were diluted to final concentrations of 1X (filled), 0.1X (grey), and 0.01X (empty). The relative cell viability was determined by MTT assay and normalised to that of the untreated Caco-2 control group. $n = 2$ independent experiments. The numerical relative cell viability data are shown in a colour-graded matrix. Bottom: the coloured bar corresponds to the relative Caco-2 viability. All data were presented as mean values with error bars representing SEMs of triplicates ($n = 3$ independent experiments), unless stated otherwise. Source data are provided as a Source Data file.

antimicrobial sensitivity and intestinal viability. We assessed the clinical symptoms, mortality and weight of the CDI model mice as previously described[35,37,38]. Challenge with *C. difficile* ($10^7$ CFU) was performed on day 0. The mortality, weight, and clinical symptoms of the mice were then monitored over the course of

9 days. The clinical symptoms were scored according to a previously established standard[38].

Figure 5 D shows that EcN-Cbh led to a 100% survival rate in the model mice, while all the controls resulted in significantly lower survival rates ranging from 60 to 14.3%. Figure 5E indicates

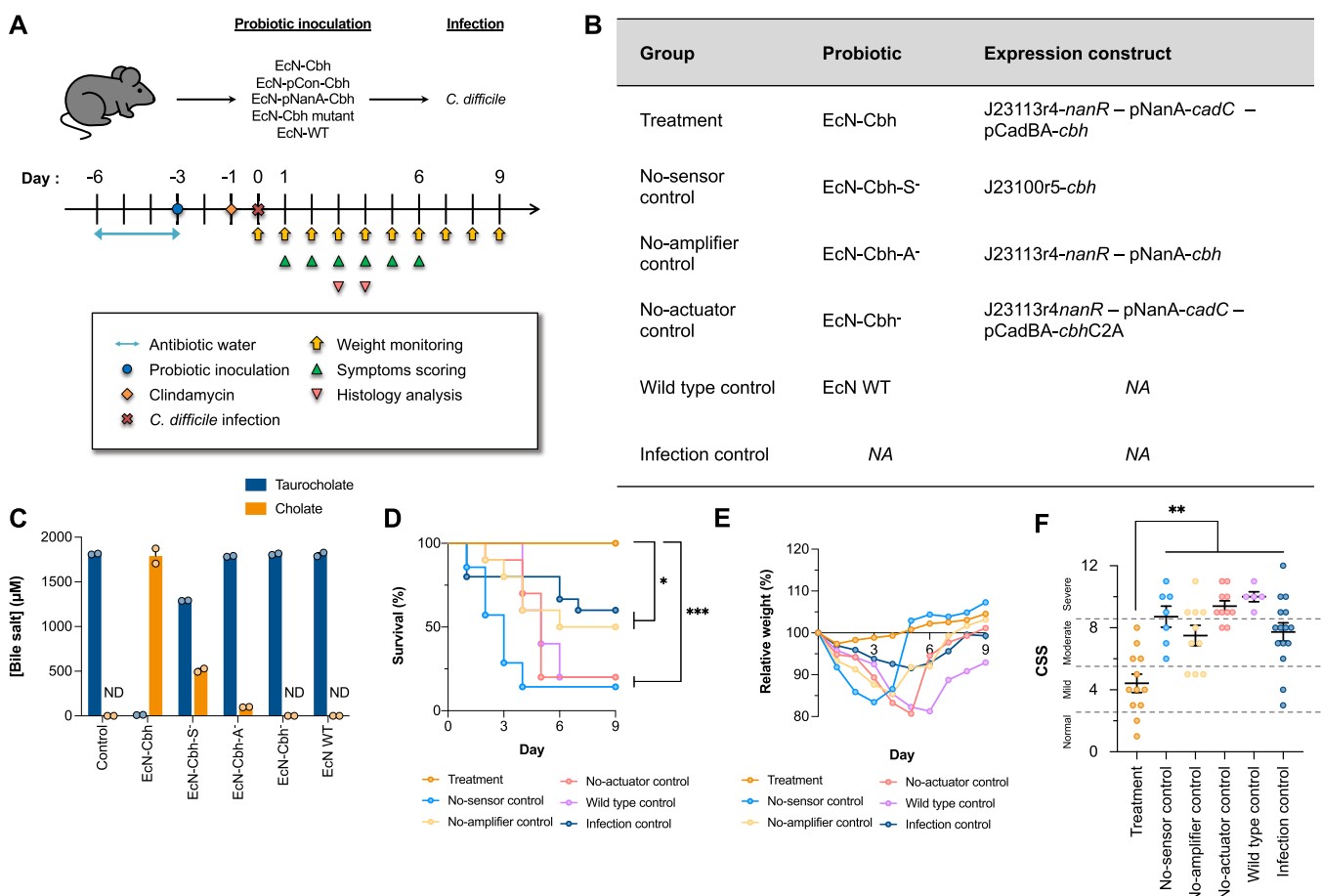

**Fig. 5 The engineered probiotics improve infection prognosis and outcomes of CDI in a murine model. A** Animal experiment timeline. **B** Table of the groups used in this study and the constructs expressed in the probiotics. An additional group for infection control in which the animals were given sucrose instead of probiotic was also used. The growth and expression characteristics of the probiotics are presented in Supplementary Fig. 5. **C** Quantification of taurocholate and cholate after incubation of the probiotics with taurocholate and sialic acid. A control reaction without EcN ("*Control*") was set up. ND not detected. The error bars represent the SEMs of results from two independent experiments. **D** Survival curves of the groups following *C. difficile* challenge. The log-rank Mantel–Cox test was performed to compare the treatment and control groups (*p* value against no-sensor control group <0.0001; no-amplifier control group = 0.0058; no-actuator control group = 0.0001; wild-type control group = 0.0002; infection control group = 0.0153). **P* < 0.05; ****P* < 0.001. **E** Relative mean weight of animals following *C. difficile* challenge. Animals were removed from the calculation on the day following their death. **F** The Clinical Sickness Score (CSS) of each animal was recorded as the highest score attained by the individual animal within 6 days postinfection. Daily CSSs are shown in Supplementary Fig. 7. The infection severity based on the score is indicated by dotted lines. Dunnett's multiple comparisons test was performed to compare the treatment and control groups (*p* value against no-sensor control group = 0.0004; no-amplifier control group = 0.0015; no-actuator control group <0.0001; wild-type control group <0.0001; infection control group = 0.0010). ***P* < 0.01. The number of independent samples for each group used are: treatment group, *n* = 12; no-sensor control group, *n* = 7; no-amplifier control group, *n* = 10; no-actuator control group, *n* = 10; wild-type control group, *n* = 5; and infection control group, *n* = 15. Source data are provided as a Source Data file.

that the infection model mice fed EcN-Cbh exhibited the least weight loss, especially between day 2 and day 4, when the model mice showed the most severe symptoms. Figure 5F shows that the group of model mice fed EcN-Cbh had the lowest clinical symptom score (CSS), 4.42. CSSs were used to indicate the severity of the infection and ranged from normal (0 to 2) to mild (3 to 5), moderate (6 to 8) or severe (9 to 12) based on a stool, behaviour and weight loss[38]. All the controls exhibited significantly higher CSSs than the EcN-Cbh-fed model mice, with mean scores ranging from 7.73 to 10.00. Finally, we performed 16 S rRNA metagenomic sequencing on a microbiome library extracted from the faeces of the animals in all the groups. Figure 6A shows that only EcN-Cbh led to a decrease in the abundance of *C. difficile* in the model mice. Figure 6B shows that among the groups that expressed Cbh, EcN-Cbh and the no-amplifier control, both of which harboured the sensor, improved the diversity of the microbiome of the model mice, as indicated by

a significant increase in the Shannon diversity index. These results together suggest that EcN-Cbh significantly reduced CDI in the model mice, as evidenced by a 100% survival rate, improved clinical symptoms and a decrease in the abundance of *C. difficile*.

**The engineered probiotics ameliorate histopathological injury in model mice**. To examine the extent of the injury caused by CDI in the model mice, we conducted a histopathological assessment of colon tissues harvested from the model mice on days 3 and 4 following death or euthanasia. Figure 6C shows the histologic injury scores (HISs) of the harvested colon tissue. The HISs were used to indicate the severity of the infection and ranged from normal (0 to 1) to mild (2 to 3), moderate (4 to 6) or severe (7 to 9) based on epithelial tissue damage, mucosal oedema and neutrophil infiltration[38] (Fig. 6D). Model mice treated with

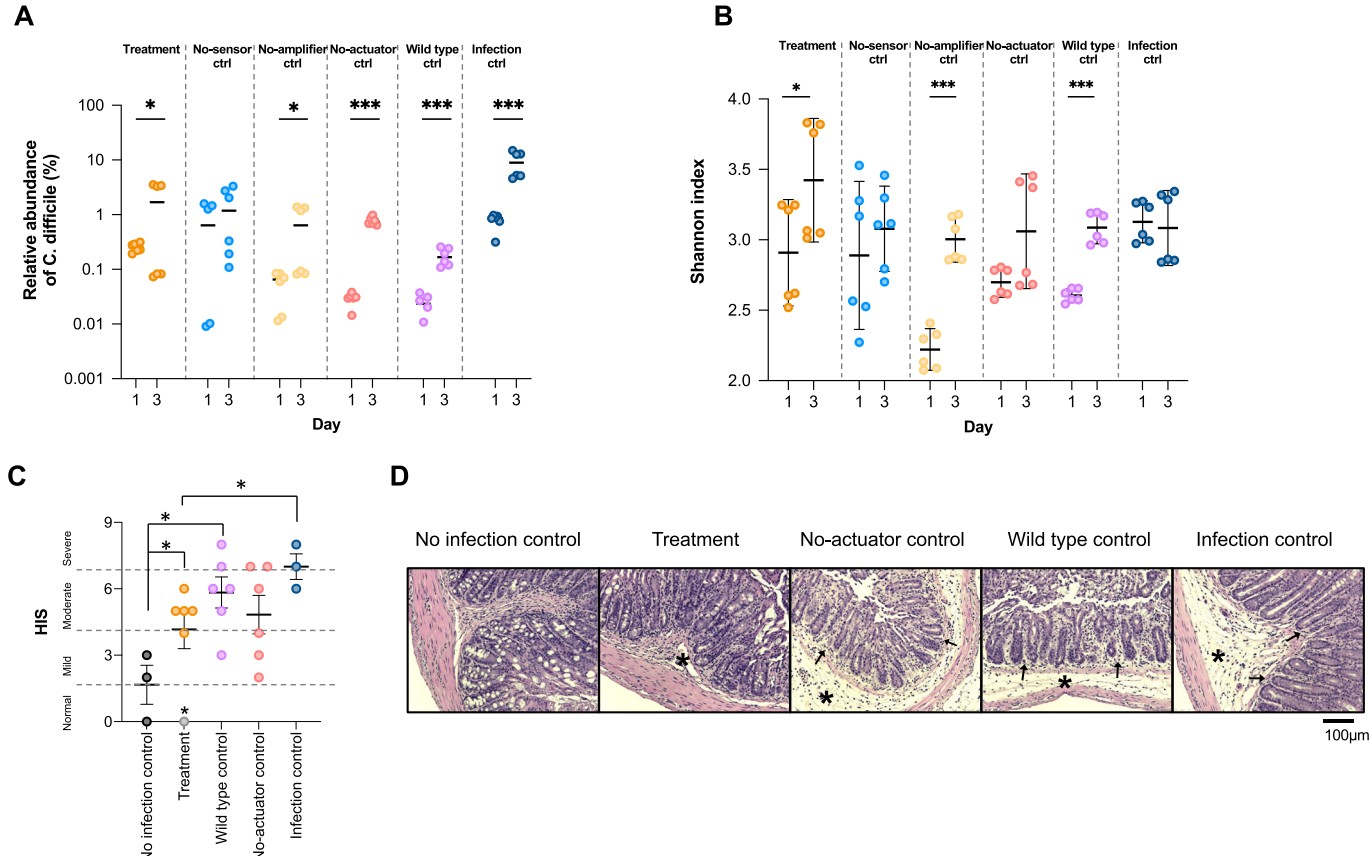

**Fig. 6 The engineered probiotics improve infection prognosis and outcomes of CDI in a murine model. A** Relative abundance of *C. difficile* from metagenomic sequencing of 16 S rRNA from a genomic library of microbiomes extracted from faecal samples of all treatment and control groups. Faecal samples from day 1 and day 3 postinfection were sequenced. Unpaired two-sided Student *t*-test was performed to compare relative abundance of *C. difficile* between day 1 and day 3 (*p* value for treatment group = 0.05, no-sensor control group = 0.2979; no-amplifier control group = 0.0412; no-actuator control group <0.0001; wild-type control group = 0.0002; infection control group = 0.0014). *P < 0.05; ***P < 0.001. Results of two independent samples for each group are used for each respective day, with three technical replicates. **B** Shannon index (alpha diversity indicator). Unpaired Student *t-test* was performed to compare the Shannon index between day 1 and day 3. *P < 0.05; **P < 0.01; ***P < 0.005. Results of two independent samples for each group are used for each respective day, with three technical replicates. **C** Histological Injury Scores (HISs) of colon tissues were assessed in a blind fashion. The infection severity based on the score is indicated by dotted lines. Mann–Whitney test was performed (outlier (*) was excluded for analysis). *P < 0.05 (*p* value for no-infection control and treatment = 0.036; no-infection and wild-type = 0.024; infection and treatment = 0.036). The numbers of independent samples for each group are no-infection control group, *n* = 3; for all other groups, *n* = 6. **D** Representative microscopic images of H&E-stained colon tissue from each group were assessed. Submucosal oedema (*) and neutrophil infiltration (arrow) were identified. The scale bar corresponds to 100 μm. Source data are provided as a Source Data file, and the black bars indicate the means and SEMs of the groups.

EcN-Cbh displayed lower HISs than the control mice. These results suggest that EcN-Cbh treatment markedly alleviates tissue damage due to CDI in model mice.

**The engineered probiotics modulate bile salt composition**. Next, to determine whether and how much the sialic acid level was elevated and to assess whether EcN-Cbh modulated bile salt profiles in vivo as hypothesised, we analysed the faeces of infection model mice as described in Fig. 7A. Figure 7B shows that the level of sialic acid was increased 6-fold upon antibiotic treatment from day -6 to -3, when EcN-Cbh was introduced, supporting our use of a sialic acid biosensor in the probiotics. Figure 7C shows that EcN-Cbh decreased the taurocholate level and increased the cholate level in the infection model mice from day -3 to day 0 before *C. difficile* was introduced. Figure 7D shows that the EcN-Cbh and the no-sensor control resulted in significantly reduced taurocholate and correspondingly elevated cholate levels. We also quantified other bile salts, glycocholate, chenodeoxycholate, lithocholate, and deoxycholate, in the faeces of the model mice

upon administration of the engineered probiotics (Supplementary Fig. 9). Supplementary Fig. 9 shows that EcN-Cbh and the no-sensor control led to an increased level of deoxycholate, a derivative of cholate, while the Cbh-expressing probiotics (i.e. EcN-Cbh, the no-sensor control, and the no-amplifier control) resulted in an increased level of chenodeoxycholate and its derivative lithocholate. These results suggest that EcN-Cbh increases cholate levels and decreases taurocholate levels in model mice, as hypothesised.

**Discussion**

We report the engineering of probiotics to inhibit CDI through dynamic regulation of bile salt hydrolase. The engineered probiotics harbour a genetic circuit that comprises a sensor, an amplifier, and an actuator. The circuit was designed to respond to and control intestinal signals to inhibit the germination and growth of *C. difficile* through two actions: inhibition of germination via hydrolase-mediated deconjugation and inhibition of vegetative growth via the deconjugated product cholate. This

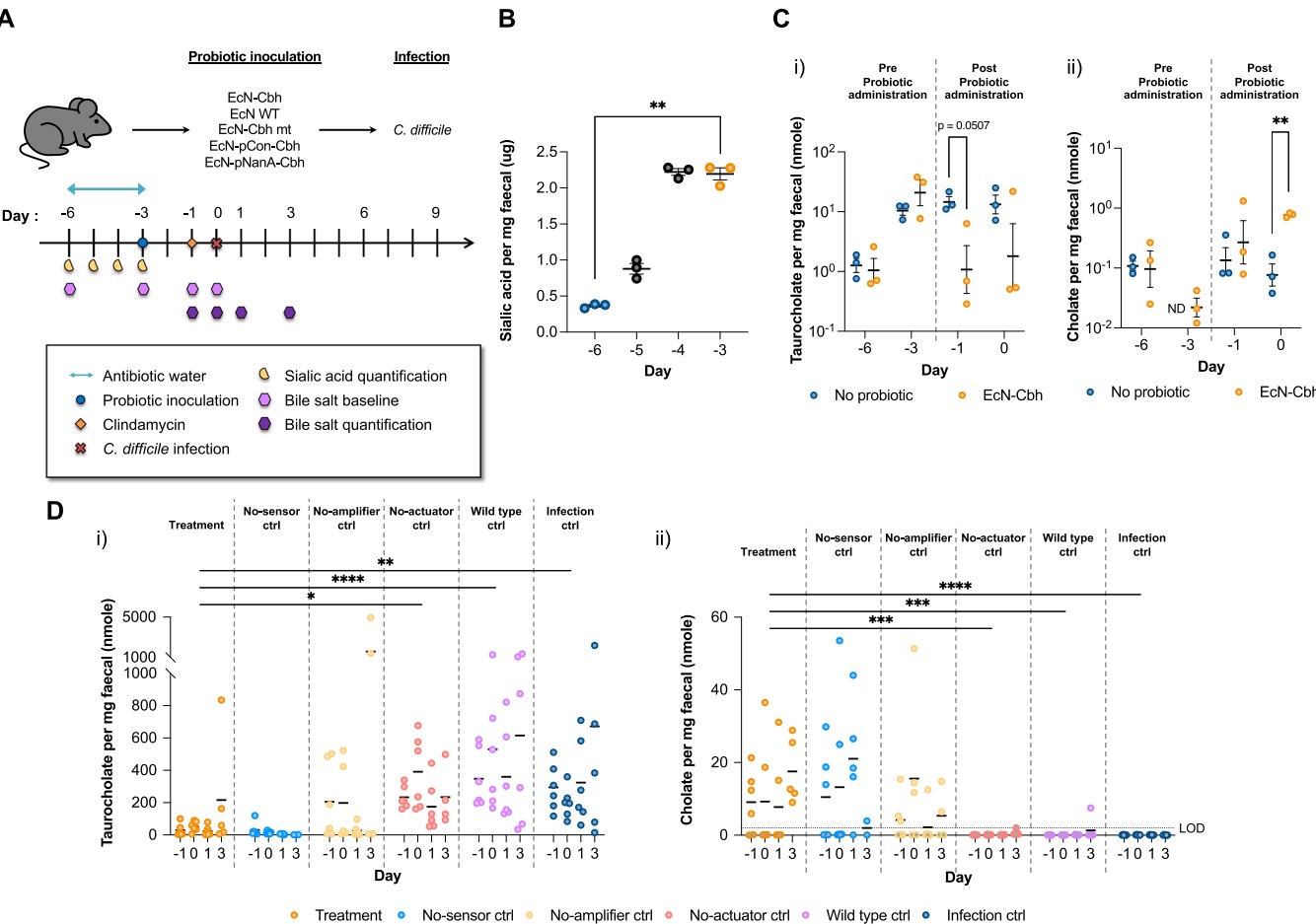

**Fig. 7 The engineered probiotics modify bile salt composition in the host gut. A** Timeline of sample collection for metabolite examination in relation to the animal experiment timeline. **B** Quantification of sialic acid in faecal samples during antibiotic-containing water treatment. Paired Student's *t*-test was performed to compare the day −6 and day -3 data ($p = 0.0027$). **$P < 0.01$. The black bars indicate the means and SEMs of the groups. $n = 3$ independent samples. **C** Quantification of (i) taurocholate and (ii) cholate in faecal samples prior to *C. difficile* challenge from day −6 to 0 in the EcN-Cbh and infection control groups ("*No probiotic*"). ND not detected. Unpaired Student's *t*-test was performed between the EcN-Cbh and infection control groups for taurocholate on day −1 ($p = 0.0507$) and cholate on day −3 ($p = 0.0056$). **$P < 0.01$. The black bars indicate the means and SEMs of the groups. $n = 3$ independent samples. **D** Quantification of (i) taurocholate and (ii) cholate in faecal samples collected from day −1 to day 3 from each group. Mixed-model ANOVA was performed to compare groups (p value for taurocholate for no-sensor control group = 0.1362; no-amplifier control group = 0.1117; no-actuator control group = 0.0152; wild-type control group <0.0001; infection control group = 0.0051; for cholate for no-sensor control group = 0.8709; no-amplifier control group = 0.2829; no-actuator control group = 0.0002; wild-type control group = 0.0001; infection control group <0.0001). *$P < 0.05$; **$P < 0.01$; ***$P < 0.001$; ****$P < 0.0001$. The black bars indicate the means and SEMs of the groups and days. LOD limit of detection. Source data are provided as a Source Data file.

work presents a targeted microbiome-modulation strategy that harnesses the host–pathogen microenvironment as the intervention target to limit the pathogenesis of infection. We based this strategy for CDI on several findings from prior studies. First, the intestinal microbiome modifies bile salt profiles[9,15–17]. Second, microbiome dysbiosis caused by antibiotic treatment disrupts intestinal bile salt metabolism[6,7,13]. Third, the dysregulation of microbiome-mediated bile salt metabolism results in an imbalance in the profile of bile salts, contributing to CDI and rCDI[7,9,12]. This imbalance in the host–pathogen microenvironment was the primary intervention target for our work, in which probiotics were genetically modified to aid in the restoration of intestinal bile salt metabolism in response to antibiotic-induced microbiome dysbiosis.

Our work suggests that modulation of bile salt metabolism can serve as a mechanism of action for the development of therapeutics against CDI, supporting prior propositions of the use of a germination/sporulation-based strategy for CDI treatment[17,18,37]. We

have provided evidence that the dynamic expression of bile salt hydrolase can significantly inhibit CDI in vivo. We hypothesise that this inhibition is aided by the rebuilding of intestinal bile salt metabolism, which promotes the prevention of spore germination and the restoration of microbiome colonisation resistance[9,17]. Recent clinical practice guidelines for CDI recommend faecal microbiota transplantation (FMT)[39], which facilitates the restoration of the pre-morbid state of the microbiome, for patients with rCDI[25,40,41]. This recommendation signifies the importance of therapeutic mechanisms that mediate the repopulation of the normal colonic microbiome for CDI treatment. Our work substantiates the value of the regulation of bile salt metabolism among such therapeutic mechanisms.

The in vitro and in vivo efficacy of the engineered probiotics demonstrated in this work suggests the potential utility of several therapeutic regimens. First, the engineered probiotics can serve as adjuvants to standard antibiotic therapy against CDI, assisting antibiotics in inhibiting and killing vegetative, virulent *C. difficile*

cells while preventing further germination of *C. difficile* spores. Notably, our engineered probiotics, which are gram-negative bacteria, can withstand the standard CDI antibiotic therapy that includes vancomycin, therefore enabling co-administration. This combination therapy may enhance the efficacy of antibiotic therapy against CDI, which disrupts the intestinal microbiome and bile salt metabolism. Second, the engineered probiotics can be administered to patients at high risk who are to undergo antibiotic therapy to prevent the potential onset of *C. difficile* infection. This probiotic administration could continue during and following antibiotic therapy, enabling the normal microbiome to repopulate the intestine, and offering protection against CDI.

It is of particular interest that the complete circuit, which comprises the sensor, amplifier, and actuator, was required for significant infection reduction (i.e., a 100% survival rate and improvement of clinical symptoms) in the model mice in the current study. From this result, we make several inferences. First, sensor-controlled regulation of bile salt hydrolase was advantageous to infection control. The genetically encoded sensor developed in our work was shown in vitro to respond dynamically to sialic acid, a proxy signal for microbiome dysbiosis whose levels were elevated upon antibiotic treatment in our murine model. We conjecture that the levels and spatial distribution of sialic acid in the model mice dynamically controlled the extent of the modulation by the hydrolase-harbouring actuator of the circuit in a spatial-specific manner. This dynamic and spatial-specific modulation might have been augmented by the sensor element (pNanA) and the amplifier element (cadC), where additional regulation by glucose and pH was applied, as stated above. The critical role of the sensor in the circuit was also evidenced by the result that the no-sensor control led to the lowest survival rate despite the constitutive expression of hydrolase, which might have imposed an additional metabolic burden on the probiotic host. This critical role is corroborated by the result that the sensor was required for hydrolase-expressing probiotics to increase the microbiome diversity in the model mice upon CDI. Second, our finding that all the control groups except the no-amplifier control group exhibited survival rates significantly lower than that of the infection control supports prior findings that probiotic administration can impair gut microbiome recovery by prolonging microbiome dysbiosis[42]. Third, both the adequate expression level of hydrolase and its controlled regulation are significant factors in ensuring the efficacy of our engineered probiotics against CDI. Only the complete circuit, which provided both the amplified and sensor-controlled expression of hydrolase, improved the survival rate and clinical symptoms and decreased the abundance of *C. difficile* in the model mice.

This study has several limitations. First, although hydrolase expression and the taurocholate-to-cholate conversion was strongly correlated in the model mice, neither was linearly correlated with the extent of infection inhibition except in the complete circuit group, which showed the most improved survival rate and clinical symptoms. For instance, the no-sensor control, which constitutively expressed hydrolase, led to a taurocholate-to-cholate conversion comparable to that of the complete circuit but resulted in a low survival rate and a high CSS. Second, the finding that the complete circuit was a prerequisite for CDI protection indicates the need for a next-level understanding of the mechanisms of the protection and the interplay between the modules of the circuit for clinical translation. This mechanistic elucidation also entails the testing of the circuits for other *C. difficile* strains, in particular, in the spore form. Third, it is plausible that hydrolase-mediated deconjugation by our engineered probiotics shifted the profiles of other intestinal metabolites, which might have helped limit the infection. Future research could examine a complete set of intestinal metabolites and microbes that contribute to anti-germination and infection inhibition as well as determine the appropriate dosage and duration for the engineered probiotics.

In conclusion, we propose that the mechanism of action of the antimicrobial strategy presented in this work, which modulates the host–pathogen microenvironment for infection control rather than exerting direct lethality, should be considered for the development of future microbiome-based antimicrobial therapeutics.

## Methods

**Culture and maintenance**. All *E. coli* strains were maintained in Luria broth (LB) medium at 37 °C with appropriate antibiotics and/or supplements unless stated otherwise (final concentrations: 30 mg/L kanamycin and 50 mg/L D-alanine). Cloning was performed with *E. coli* Top10 (Invitrogen), heterologous protein expression was performed with *E. coli* BL21 (Thermo Fisher Scientific), and characterisation was performed with a modified *E. coli* Nissle 1917 EcN strain[22]. *C. difficile* culture was carried out in a Coy Lab Vinyl Anaerobic Chamber and anaerobic jar to maintain anaerobic conditions. Cells were cultured in brain heart infusion-supplemented (BHIS) medium supplemented with 5% *w/v* yeast extract and 0.03% *w/v* L-cysteine. Taurocholate was supplemented as necessary. The cells were incubated anaerobically at 37 °C. *C. difficile* CD630, VPI10463, BAA1870 and 9689 were obtained from the American Type Culture Collection (ATCC). The Caco-2 cell line (ATCC) was maintained in Dulbecco's modified Eagle's medium (DMEM) supplemented with 20% foetal bovine serum and 1% penicillin/streptomycin (10000 U/mL). The cells were incubated at 37 °C under a 5% $CO_2$/95% air atmosphere. Cell passage was performed at ~70 to 90% confluence.

**_C. difficile_ endospore germination efficiency assay and vegetative cell growth assay**. *C. difficile* endospores were extracted by resuspending *C. difficile* CD630 following 6 days of incubation in 50% *v/v* ethanol/PBS for 1 h. The cells were then washed twice in PBS before incubation at 70 °C for 20 min and resuspended in PBS with 1% *w/v* bovine serum albumin (BSA). The extracted endospores were counted under anaerobic conditions for CFU determination by tenfold serial dilution on BHIS agar containing 2 mM taurocholate, unless otherwise stated[13]. The endospores were diluted in PBS with 1% *w/v* BSA to a final concentration of ~$10^6$/mL and then counted under anaerobic conditions for CFU determination on BHIS plates containing bile salts at concentrations ranging from 1 to 5 mM. To assess endospore germination efficiency or vegetative cell growth with Cbh-treated or engineered EcN-treated taurocholate, purified 10 μM Cbh or engineered EcN at an optical density at 600 nm wavelength ($OD_{600}$) of 0.5 was incubated with 10 mM taurocholate for 3 h before the addition of *C. difficile* endospores or vegetative cells. The bile salt reaction mixture was diluted five times to achieve a final 2 mM bile salt concentration and incubated anaerobically with endospores at a final concentration of ~$10^6$/mL for 1 h or with vegetative *C. difficile* at an $OD_{600}$ of 0.2 (~$2 \times 10^7$ CFU/mL) for 12 h. *C. difficile* were then counted for CFU determination under anaerobic conditions on BHIS agar. All assays were performed in triplicate unless stated otherwise.

**Molecular cloning and protein expression**. The BglBrick standard was used in the design and cloning of biological parts[43]. Cloning was performed with a pEaaK vector[22]. The Cbh gene was codon-optimised for expression in *E. coli* and synthesised with the addition of a C-terminal hexahistidine tag (IDT Technologies). Cbh was expressed and purified through immobilised metal ion affinity chromatography (IMAC) with nickel-charged agarose resins followed by size exclusion chromatography on a HiLoad 16/600 Superdex 75 pg column (GE Healthcare Life Sciences). Selected fractions were then concentrated with ultrafiltration columns at a 10 kDa molecular weight cut-off (MWCO) in protein buffer (50 mM Tris-HCl, 100 mM NaCl, 10% *v/v* glycerol, pH 8.0).

**Characterisation of bile salt deconjugation activity**. Bile salts were detected with a high-performance liquid chromatography (HPLC) system. To assay protein activity, 10 μM purified Cbh was incubated with 10 mM taurocholate in protein buffer at 37 °C for 3 h. Four volumes of methanol were then added to precipitate the proteins. The mixture was dried by a vacuum concentrator, resuspended in 70% *v/v* acetonitrile, and subjected to HPLC analysis. For assays involving probiotics, the cells were pre-induced with 0.2% *w/v* sialic acid for 3 h at 37 °C, adjusted to an $OD_{600}$ of 3.0 and further incubated with 10 mM taurocholate for 3 h. The supernatants were then collected for quantification of the bile acids using UPLC-electrospray ionisation (ESI)-tandem MS (MS/MS) in negative ionisation mode with an Eclipse Plus C18 column (2.1 mm × 100 mm, 1.8 μm particle size) (Agilent Technologies). (See 'Liquid chromatography (LC)-mass spectrometry (MS) quantification of faecal bile salts' for further details).

**Growth and protein expression assay**. Cells or constructs were characterised with growth assays or fluorescent protein expression assays on a Synergy H1 Multimode plate reader (BioTek). All assays were performed in triplicate unless stated otherwise. The absorbance of the cell culture at a 600 nm wavelength ($OD_{600}$) was read to indicate the cell density. Green fluorescent protein (GFP)

fluorescence (excitation: 485 nm, emission: 528 nm) in the sample medium relative to the blank medium was read to indicate GFP expression. The initial cell densities ($OD_{600}$ values) of pre-cultures at the early exponential phase were adjusted to an absorbance of 0.2 prior to microtitre plate reading. Then, 200 μL of culture was added to each well of a clear-bottom 96-well microtitre plate along with a blank medium for background reading normalisation and water for pathlength normalisation. Appropriate chemicals were added accordingly for induction of cells (final concentrations: 0.2% w/v sialic acid, 0.2% w/v L-arabinose, and 20 mM glucose). The microtitre plates were incubated at 37 °C and agitated at 225 rpm with orbital shaking in the plate reader. Cbh expression was confirmed in an immunoblot using an anti-His tag antibody conjugated with HRP at 1:1000 dilution (#9991 S, Cell Signaling Technology).

**Flow cytometry characterisation of protein expression.** Fluorescent protein expression under multiple inducer concentrations was characterised on an Accuri C6 flow cytometer (Becton Dickinson). The initial cell densities ($OD_{600}$ values) of pre-cultures at the early exponential phase were adjusted to an absorbance of 0.2. Then, 200 μL of culture was added to each well of a 96-well microtitre plate. Appropriate chemicals were added accordingly for induction of cells (final concentrations: 0.05 to 0.2% w/v sialic acid and 0.00001 to 0.1% w/v L-arabinose). The cultures were incubated at 37 °C for 12 h with agitation at 1000 rpm. The cells were then diluted 100 times in water and sampled with a flow cytometer. Ten thousand size-gated samples were taken for each experiment, and the median fluorescence reading was quantified.

***C. difficile* toxin extraction and coculture assay.** Engineered EcN-treated bile salt supernatants were collected as previously described. The supernatants were diluted five times with BHIS and then incubated anaerobically with *C. difficile* endospores ($10^6$/mL) at 37 °C. Supernatants of the coculture were collected at regular intervals and concentrated ten times using speedvac vacuum concentrator (Savant Speed-Vac™, Thermo Fisher Scientific), and the buffer was exchanged for protein buffer (50 mM Tris-HCl, 100 mM NaCl, pH 8.0) with 50 kDa MWCO ultrafiltration columns. The concentrated supernatants were probed with 500 μg/L anti-TcdA antibody (PCG4, ab19953, Abcam) and anti-mouse IgG conjugated to HRP at 1:2000 dilution (#7076, Cell Signaling Technology) in an immunoblot assay. Direct coculture of Caco-2 cells and *C. difficile* was not possible due to the differences in laboratory growth conditions. An indirect coculture assay was performed by incubating concentrated supernatants from *C. difficile* with Caco-2 cells. Caco-2 cells with passage numbers between 16 and 20 were used for the assay. Cells ($5 \times 10^4$/mL) in 100 μL of culture medium were seeded in each well of clear-bottom 96-well microtitre plates. The cells were allowed to proliferate for 48 h at 37 °C. Fresh complete medium without antibiotics was provided, and concentrated supernatants were added to the Caco-2 cells. The supernatant was concentrated tenfold and then dilute ten times in Caco-2 growth media to achieve a final concentration equivalent to one time of the *C. difficile* culture during which sufficient growth medium required for Caco-2 growth was maintained. The tenfold-concentrated supernatants were diluted to final concentrations of 1-fold, 0.1-fold or 0.01-fold in the cell medium in duplicate. The cells were further incubated for 48 h before viability assessment with a methylthiazolyldiphenyl-tetrazolium bromide (MTT) assay. The $OD_{570}$ value (formazan signal) and the $OD_{630}$ value (reference signal) were read on a plate reader. The relative cell viability compared to that of untreated Caco-2 cells is reported.

**CDI animal models.** All procedures were conducted under Institutional Animal Care and Use Committee (IACUC) guidelines and in conformity with protocols approved by the NUS IACUC (R18–0329) and all relevant ethical regulations. The CDI mouse model has been previously reported[34]. Male C57BL/6 mice (5–6 weeks old) were provided with an antibiotic cocktail in water (0.4 mg/mL kanamycin, 0.035 mg/mL gentamicin, 850 U/mL colistin, 0.215 mg/mL metronidazole, 0.045 mg/mL vancomycin) for 3 days following acclimatisation (day -6 to -3; pre-infection). A dose of clindamycin (10 mg/kg) was administered intraperitoneally on day −1. Infection with *C. difficile* VPI10463 ($10^7$ CFU of vegetative cells) was performed on day 0 through oral gavage. The mice were maintained on a complex carbohydrate-free diet (D15091702Bi AIN-93G, Research Inc.). Probiotics ($10^9$ CFU) were provided through oral gavage on day −3 as necessary. The animals were monitored up to day 9 postinfection, under standard housing conditions (12 light/12 dark cycle, 22–24 °C with humidity set at 40–50%). Symptoms were scored with a standardised CSS system as described previously[38]. Briefly, the scoring was based on three major symptoms observed in humans: behavioural changes, stool characteristics and weight loss. Each category of the CSS system was scored from 0 to 4, and individual scores were added to provide an overall severity score. Kaplan-Meier survival curves were generated and analysed with the log-rank test using GraphPad Prism.

**Microbiome sequencing and analysis.** Metagenomic DNA was extracted from faecal samples with the Zymobiomics DNA miniprep kit (Zymo Research). Metagenomic libraries were prepared by amplifying the V3-V4 region of prokaryotes 16 s rRNA[44], and then indexed with Illumina XT Index Kit v2. The purified indexed libraries were quantified using KAPA Library Quantification Kit (Roche) and the quality of the amplicons was measured in an Agilent Tapestation (Agilent Technologies). The pooled libraries were sequenced in an Illumina Miseq platform

with Miseq Reagent Kit v2. Sequencing reads were demultiplexed and were analysed using the QIIME2[45] workflow for 16 S amplicons. the demultiplexed reads were quality-checked, denoised and filtered for chimera to output the representative sequences and OTU table. The Greengenes database (v13_8) was used as a training set for the taxonomy assignment of the OTUs[46]. Microbiome 16 s rRNA sequencing data generated in this study have been deposited in the Sequence Read Archive (SRA) under accession code PRJNA844050.

**Histopathological analysis of animal tissue.** Colon tissues were collected from animals, fixed with 10% formalin solution, and then processed for paraffin embedding. Haematoxylin and eosin (H&E) staining was performed on sectioned slides, and then the HIS was determined according to a standardised protocol[38] by a pathologist who was blinded to the treatment information. Images were captured on Leica DMi8 Inverted Microscope.

**Derivation of sialic acid for HPLC detection.** Faecal samples were collected and homogenised in water. The homogenised samples were filtered and resuspended in 7 mM 1,2-diamino-4,5-methylenedioxybenzene dihydrochloride (DMB), 0.75 M β-mercaptoethanol and 18 mM sodium hydrosulfite in 1.4 M acetic acid. The samples were incubated in the dark at 50 °C for 2.5 h and then analysed by reverse-phase HPLC with an octadecyl silica Inertsil ODS-3 column (4.6 mm × 250 mm, 5 μm particle size) (GL Sciences). A 5 μL aliquot was injected into the column. The run was performed at a flow rate of 0.9 mL/min with linear gradient elution at 5:88:7 to 20:73:7 v/v/v acetonitrile:water:methanol solvent until a run time of 15 min, isocratic elution at 20:73:7 until a run time of 18 min, gradient elution to 80:13:7 until a run time of 19 min, isocratic elution at 80:13:7 until a run time of 22 min, gradient elution to 5:88:7 until a run time of 23 min, and final recalibration at 5:88:7 for 3 min. The derivatised samples were detected under 373 nm excitation and 448 nm emission with a fluorescence detector. Sialic acid was eluted at approximately 13.4 min. Standard curves were constructed based on the areas of the corresponding elution peaks in the elution spectra.

**Liquid chromatography (LC)-mass spectrometry (MS) quantification of faecal bile salts.** Faecal samples were collected and homogenised in Zymobiomics DNA miniprep lysis buffer (Zymo Research). Ten microlitres of the homogenised mixture was mixed with 90 uL of LC-MS grade methanol and filtered through a 0.2 μm syringe PTFE filter for organic solutions (Millipore). Quantification of the bile acids was performed in a UPLC-electrospray ionisation (ESI)-tandem MS (MS/MS) in negative ionisation mode with an Eclipse Plus C18 column (2.1 mm × 100 mm, 1.8 μm particle size) (Agilent Technologies). Ten microlitres of the samples were injected into the column with mobile phases A and B consisting of 0.1% formic acid in 7.5 mM ammonium acetate in water and 7.5 mM ammonium acetate in acetonitrile, respectively. The separation of the analytes was performed at a flow rate of 0.25 mL/min with a gradient at 40%B from 0 to 5 min, 100%B from 5 to 7 min and back to 40%B for 3 min. The gas temperature was set to 250 °C at a flow of 15 L/min, and the sheath temperature to 400 °C at 12 L/min. Capillary and nozzle voltages were set at 3500 and 2000 V, respectively. The bile acids were detected using the following multiple reaction monitoring conditions: cholic acid (m/z 407→341, collision energy 36 V), chenodeoxycholic acid (m/z 391→391, CE 30 V); deoxycholic acid (m/z 391→345, CE 36 V); glycocholic acid (m/z 464→74, CE 36 V), lithocholic acid (m/z 375→375, CE 30 V) and taurocholic acid (m/z 514→80, CE 64 V). The gas temperature was set to 250 °C at a flow of 15 L/min, and the sheath temperature to 400 °C at 12 L/min. Mass spectrometry data will be provided upon request.

**Statistics and reproducibility.** All in vitro experiments conducted on 96-well microtitre plates, with the exception of the flow cytometry and MTT assays, were performed in triplicate. All in vitro CFU and HPLC experiments were performed in triplicates unless stated otherwise. Immunoblot assays were repeated independently thrice and representative blots were shown. Sample sizes for the animal study were selected based on a previous study[35]. Histology samples were determined by a pathologist who was blinded to the treatment information. One image was captured for each histology slide and the representative was selected. All reported error bars represent the standard error of the mean (SEM) values unless stated otherwise. The student's t-test was performed for statistical analysis of bile salt quantities, germination assay results, growth, and metabolite quantities. Standard curves for HPLC and LC-MS analysis were generated by the linear regression method. The log-rank Mantel–Cox test was performed to analyse animal survival, Dunnett's multiple comparisons test was performed to analyse the animal CSS, and mixed-model ANOVA was performed to analyse the taurocholate and cholate quantification. Microsoft Excel, Graphpad Prism and SPSS were used for data analysis.

**Reporting Summary.** Further information on research design is available in the Nature Research Reporting Summary linked to this article.

# Data availability

Microbiome 16 s rRNA sequencing data that support the findings of this study have been deposited in the Sequence Read Archive (SRA) under accession code PRJNA844050. All

other data supporting the findings of this study are available within the article and its supplementary information files. Source data are provided with this paper.

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

## Acknowledgements

We thank Dr. R. Ravisankar for his assistance in the histopathological assessment. This work was supported by the Synthetic Biology Initiative of the National University of Singapore (DPRT/943/09/14, M.W.C.), the Summit Research Programme of the National University Health System (NUHSRO/2016/053/SRP/05, M.W.C.), NUS Medicine Synthetic Biology Translational Research Programme (NUHSRO/2020/077/MSC/02/SB, M.W.C.), Investigatorship of the National Research Foundation of Singapore (NRF-NRFI05-2019-0004, M.W.C.), ISF-NRF Joint Programme of the National Research Foundation of Singapore (NRF2019-NRF-ISF003-3208, I.Y.H.), the Ministry of Education of Singapore (NUHSRO/2020/046/T1/3, M.W.C.) and the U.S. Air Force Office of Scientific Research—Asian Office of Aerospace Research and Development (FA2386-18-1-4058, M.W.C.). This work used the resources of the Singapore BioFoundry, a bio-manufacturing research facility located at the National University of Singapore.

## Author contributions

E.K., I.Y.H. and M.W.C. conceived, designed and performed the study and wrote the manuscript. E.K., I.Y.H., H.L.L, R.D.S., J.W.J.L., Y.S.L., J.C.M. and M.W.C. analysed the data. J.W.J.L., Y.S.L. and J.C.M. edited the manuscript. M.W.C. obtained funding and supervised the study. All authors critically reviewed the manuscript and approved the final version of the manuscript for submission.

## Competing interests

E.K., I.Y.H. and M.W.C. have filed a provisional patent application (application number 17/599,998 submitted by the National University of Singapore) based on the work described in this manuscript. The remaining authors declare no competing interests.
