## [Peer Review File · Nature Communications]

Reviewers' Comments:

Reviewer #1:

Remarks to the Author:

In this resubmitted the authors Chang and colleagues address many of this reviewer's critique. However, one core critique report whether the production of cholate by EcN-cbh is the mechanism of protection in the C. difficile mouse model remains unresolved.

Original Critiques:

1. "In Figure 5, What is the bacterial burden of both C. difficile and EcN-cbh during the infection system?"

This information will provide the reader with context to understand the morbidity/mortality data and

the taurocholate/cholate ratio data in figure 5 & Figure 6. The in vivo C. difficile burden data will also

complement the in vitro data in Figure 4C. The 16S data identifying Peptostreptococcaceae in Figure

5G is insufficient in both taxonomy identity (family level) and sample size (n=2) to determine if EcN-

cbh reduces C. difficile burden in mice following infection."

- The authors have added C. difficile burden (as measured by relative abundance from shotgun metagenomic sequencing data) and increased the sample size sufficiently. However the burden of EcN-cbh during the infection system was not addressed. Suppl. Fig 6C measures EcN following antibiotic treatment without infection. This experimental data does not address this reviewer's critique. First, it is unknown which EcN strain is measured in suppl fig 6C. Second, more importantly, EcN engraftment may be different in the context of the inflammatory response following infection. The authors should show the burden of all EcN (EcN-Cbh, EcN-Cbh-S-, EcN-Cbh-A-, EcN-Cbh-, EcN WT) following inoculation in the context of C. difficile infection. The shotgun metagenomic sequencing dataset used for Fig 6G should be sufficient to use to do this analysis of EcN burden.

2. "The author examines the in vivo taurocholate and cholate levels following different probiotic administration in Figure 6. Cholate isn't the only secondary bile acid produced in the intestine capable

of inhibiting C. difficile. Did they authors profile the levels of other primary and secondary bile acids in

these groups? As the authors note in their discussion, the no sensor control has equivalent cholate to taurocholate as EcN-cbh but the no sensor E. coli does not convey protection. Perhaps other secondary bile acid levels can account for this discrepancy and provide further insight into the mechanism of protection for EcN-cbh."

- The authors have adjusted their conclusions and discussion to acknowledge that conversion to cholate is not driving host protection in the EcN-cbh group since the no sensor control also enhances cholate production and mouse still succumb. The authors conclude that the amplifier system is necessary for in vivo protection and speculate that spatial distribution of sialic acid might create micro-niches that drive protection. This is an interesting potential explanation, however it is not experimentally tested. Therefore, a main conclusion of the manuscript that the probiotic inhibits C. diff infection by "dynamic modulation of intestinal metabolites", as stated in the title is not fully substantiated. The new data in Figure 6G is not sufficient to support this claim as C. diff goes down from day 1 to 3 but there is no difference in relative abundance at day 3 between treatment groups. Comparing burden changes over time (from day 1 to day 3) is not the most appropriate comparison when using relative abundance since the total number of bacteria in feces may be changing as bacteria repopulate the intestine following antibiotic treatment.

The remainder of this reviewer's critiques have been addressed.

Reviewer #2:

Remarks to the Author:

The paper by Koh et al has addressed the comments of the previous submission thoroughly. While there are interpretations that can still be quibbled with, I am pleased with the responses and have a few comments that need to be considered prior to publication.

1. In revising figures 5 and 6, the authors note statistical significance of many of the panels in the legends, however asterisks are missing from many of the figures. I presume they were left off since if none of this data was significant I presume it wouldn't be published?

2. Figure 5I. Disease severity is not a continuous variable so a T-test is not an appropriate statistical test. Please revise.

3. Line 342. Be careful with the wording here. Since you did not have a no antibiotic control it should not be stated that you recovered diversity as it implies you are moving back to the normal state. State that it increased diversity instead.

4. The biggest point I would like to make is the discussion of the two of the results that were picked up by multiple of the reviewers. The first is the fact that although the fully engineered biosensor works well in preventing disease and mortality, all the other sensors are either worse or the same as the infection control. While addressed adequately in the discussion now, I think it also warrants a statement about how important it will be to understand this issue prior to testing in humans as most of the constructs do worse. The second is the fact that VPI was used and was administered as vegetative cells. A statement in the limitations about this fact and the need to test other *C. difficile* strains delivered as spores will be important for future mechanistic understanding. Many readers will miss this point. I don't think either of these issues detracts from the manuscript and will allow people to catch these important points that many will miss.

Reviewer #3:

Remarks to the Author:

I recommend acceptance of this revised version of the paper as the authors have addressed my prior concerns.

Revised manuscript: Point by Point Response

NCOMMS-22-06913-T, "*Engineering probiotics to inhibit Clostridioides difficile infection by dynamic modulation of intestinal metabolites*"

We thank the editor and reviewers for the helpful comments. We have incorporated as many of the reviewers' suggestions as possible into the revised manuscript.

Reviewer #1 (Remarks to the Author):

In this resubmitted the authors Chang and colleagues address many of this reviewer's critique. However, one core critique report whether the production of cholate by EcN-cbh is the mechanism of protection in the *C. difficile* mouse model remains unresolved.

Original Critiques:

1. "In Figure 5, What is the bacterial burden of both *C. difficile* and EcN-cbh during the infection system? This information will provide the reader with context to understand the morbidity/mortality data and the taurocholate/cholate ratio data in figure 5 & Figure 6. The in vivo *C. difficile* burden data will also complement the in vitro data in Figure 4C. The 16S data identifying Peptostreptococcaceae in Figure 5G is insufficient in both taxonomy identity (family level) and sample size (n=2) to determine if EcN-cbh reduces *C. difficile* burden in mice following infection."

- The authors have added *C. difficile* burden (as measured by relative abundance from shotgun metagenomic sequencing data) and increased the sample size sufficiently. However the burden of EcN-cbh during the infection system was not addressed. Suppl. Fig 6C measures EcN following antibiotic treatment without infection. This experimental data does not address this reviewer's critique. First, it is unknown which EcN strain is measured in suppl fig 6C. Second, more importantly, EcN engraftment may be different in the context of the inflammatory response following infection. The authors should show the burden of all EcN (EcN-Cbh, EcN-Cbh-S-, EcN-Cbh-A-, EcN-Cbh-, EcN WT) following inoculation in the context of *C. difficile* infection. The shotgun metagenomic sequencing dataset used for Fig 6G should be sufficient to use to do this analysis of EcN burden.

We thank the reviewer for the comment. First, we would like to clarify that the EcN strain measured in Supplementary Figure 6C is EcN-Cbh. We have revised the figure legend to make this point clear in Supplementary Figure 6 as follows:

Line 90: "Supplementary Figure 6. C) Enumeration of the engineered probiotics EcN-Cbh in faecal samples collected from mice given probiotics, after the antibiotic cocktail treatment."

*Second, following the reviewer's suggestion, we have included a new dataset showing the burden of all EcNs following inoculation in the context of *C. difficile* infection in new Supplementary Figure 8, below. We measured the burden by enumerating EcN strains in the faecal samples on Day 1 and Day 3 instead of analysing the sequencing dataset because the sequencing result did not allow us to discern EcN from other species with high similarity (e.g. other *Escherichia coli* strains, *Shigella*, etc.). The result showed that the number of viable EcN was comparable among all groups, where no significant difference was observed.*

New Supplementary Figure 8. Probiotics in mice with *C. difficile* infection. Enumeration of viable engineered or wild-type probiotics in faecal samples collected from mice on day 1 and day 3 post *C. difficile* infection.

2. "The author examines the in vivo taurocholate and cholate levels following different probiotic administration in Figure 6. Cholate isn't the only secondary bile acid produced in the intestine capable of inhibiting *C. difficile*. Did the authors profile the levels of other primary and secondary bile acids in these groups? As the authors note in their discussion, the no sensor control has equivalent cholate to taurocholate as EcN-cbh but the no sensor *E. coli* does not convey protection. Perhaps other secondary bile acid levels can account for this discrepancy and provide further insight into the mechanism of protection for EcN-cbh."

- The authors have adjusted their conclusions and discussion to acknowledge that conversion to cholate is not driving host protection in the EcN-cbh group since the no sensor control also enhances cholate production and mouse still succumb. The authors conclude that the amplifier system is necessary for in vivo protection and speculate that spatial distribution of sialic acid might create micro-niches that drive protection. This is an interesting potential explanation, however it is not experimentally tested. Therefore, a main conclusion of the manuscript that the probiotic inhibits *C. diff* infection by "dynamic modulation of intestinal metabolites", as stated in the title is not fully substantiated. The new data in Figure 6G is not sufficient to support this claim as *C. diff* goes down from day 1 to 3 but there is no difference in relative abundance at day 3 between treatment groups. Comparing burden changes over time (from day 1 to day 3) is not the most appropriate comparison when using relative abundance since the total number of bacteria in feces may be changing as bacteria repopulate the intestine following antibiotic treatment.

We thank the reviewer for the comment. Following the reviewer's suggestion, we have revised the claims that imply that metabolite modulation was the primary mechanism of protection throughout the manuscript as follows:

Line 284: "We report the engineering of probiotics to inhibit CDI through dynamic regulation of bile salt hydrolase."

Line 302: "We have provided evidence that dynamic expression of bile salt hydrolase can significantly inhibit CDI in vivo."

Line 310: "Our work substantiates the value of the regulation of bile salt metabolism among such therapeutic mechanisms."

Line 330: "First, sensor-controlled regulation of bile salt hydrolase was advantageous to infection control."

We have also revised the title of the manuscript to reflect the revised claim as follows:

Title: “Engineering probiotics to inhibit Clostridioides difficile infection by dynamic regulation of intestinal metabolism”

We have also expanded the discussion to suggest that a more thorough understanding of the mechanism of protection be required for clinical translation:

Line 358: “This study has several limitations... Second, the finding that the complete circuit was prerequisite for CDI protection indicates the need for the next-level understanding of the mechanisms of the protection and the interplay between the modules of the circuit for clinical translation. This mechanistic elucidation also entails the testing of the circuits for other C. difficile strains, in particular, in the spore form. Third,..”

As for the burden changes from Day 1 to Day 3, we would like to clarify that the following standardisation methods were applied to the analysis to ensure that the sequencing results were suitable for comparison. First, the concentration of extracted faecal DNA and subsequently amplified 16s rRNA amplicons from each sample were standardised prior to sequencing to prevent sequencing bias. Second, the resulting sequences reads were rarefied to normalize the reads for a better comparison of the diversity in each sample. Rarefaction curve analysis was also performed to ensure that the normalisation was sufficient to accurately represent the diversity in each sample. Based on these normalisations, we reasoned that the presented C. difficile relative abundance was a suitable representation of C. difficile in the samples.

The remainder of this reviewer's critiques have been addressed.

Reviewer #2 (Remarks to the Author):

The paper by Koh et al has addressed the comments of the previous submission thoroughly. While there are interpretations that can still be quibbled with, I am pleased with the responses and have a few comments that need to be considered prior to publication.

1. In revising figures 5 and 6, the authors note statistical significance of many of the panels in the legends, however asterisks are missing from many of the figures. I presume they were left off since if none of this data was significant I presume it wouldn't be published?

We thank the reviewer for the comment. We have realised that the asterisks disappeared when our manuscript was converted into PDF format. Please refer to Figures 5 and 6, below, for the statistical significance.

Figure 5: The engineered probiotics improve infection prognosis and outcomes of CDI

Figure 6: The engineered probiotics modify bile salt composition in the host gut.

2. Figure 5I. Disease severity is not a continuous variable so a T-test is not an appropriate statistical test. Please revise.

We thank the reviewer for the comment. We have re-conducted our statistical analysis by employing the Mann-Whitney test. As shown in revised Figure 5 below, statistically significant changes were observed in histological scoring between 'No-infection and Wild-type', 'No-infection and Treatment' and 'Infection and Treatment' groups.

Figure 5. I) Histological Injury Scores (HISs) of colon tissues assessed in a blind fashion. The infection severity based on the score is indicated by dotted lines and corresponds to the following scale: normal,

0 to 1; mild, 2 to 3; moderate, 4 to 6; and severe, 7 to 9. The black bars indicate the means and SEMs of the groups. Mann-Whitney test was performed (outlier (*) was excluded for analysis). *P < 0.05.

3. Line 342. Be careful with the wording here. Since you did not have a no antibiotic control it should not be stated that you recovered diversity as it implies you are moving back to the normal state. State that it increased diversity instead.

We thank the reviewer for the comment. We have revised our statement as follows:

Line 341: "This critical role is corroborated by the result that the sensor was required for hydrolase-expressing probiotics to increase the microbiome diversity in the model mice upon CDI."

4. The biggest point I would like to make is the discussion of the two of the results that were picked up by multiple of the reviewers. The first is the fact that although the fully engineered biosensor works well in preventing disease and mortality, all the other sensors are either worse or the same as the infection control. While addressed adequately in the discussion now, I think it also warrants a statement about how important it will be to understand this issue prior to testing in humans as most of the constructs do worse. The second is the fact that VPI was used and was administered as vegetative cells. A statement in the limitations about this fact and the need to test other *C. difficile* strains delivered as spores will be important for future mechanistic understanding. Many readers will miss this point. I don't think either of these issues detracts from the manuscript and will allow people to catch these important points that many will miss.

We thank the reviewer for the comments. We have added the following statement to our discussion to incorporate the reviewer's suggestion:

*Line 358: "This study has several limitations... Second, the finding that the complete circuit was prerequisite for CDI protection indicates the need for the next-level understanding of the mechanisms of the protection and the interplay between the modules of the circuit for clinical translation. This mechanistic elucidation also entails the testing of the circuits for other *C. difficile* strains, in particular, in the spore form. Third,.."*

Reviewer #3 (Remarks to the Author):

I recommend acceptance of this revised version of the paper as the authors have addressed my prior concerns.

We thank the reviewer again for the comments.